# Body Composition, Training Volume/Pattern and Injury Status of Slovenian Adolescent Female High-Performance Gymnasts

**DOI:** 10.3390/ijerph18042019

**Published:** 2021-02-19

**Authors:** Boštjan Jakše, Barbara Jakše, Ivan Čuk, Dorica Šajber

**Affiliations:** 1Department of Food Science, Biotechnical Faculty, University of Ljubljana, 1000 Ljubljana, Slovenia; 2Sole Proprietor, 1230 Domžale, Slovenia; barbara.tursic@gmail.com; 3Department of Gymnastics, Faculty of Sport, University of Ljubljana, 1000 Ljubljana, Slovenia; ivan.cuk@fsp.uni-lj.si; 4Departments of Swimming, Faculty of Sport, University of Ljubljana, 1000 Ljubljana, Slovenia; dorka.sajber@fsp.uni-lj.si

**Keywords:** gymnasts, high performance, elite-level, female, body composition, training volume, injuries

## Abstract

Body composition (BC), training volume/pattern, and injury status are a few important factors affecting training quality and sport performance in female artistic gymnastics. We aimed to examine BC status, training volume/pattern, and injury status during the first competition period. Our cross-sectional study included 17 female gymnasts (age: 17.4 ± 4.1 years, body height: 159.8 ± 6.2 cm, and body weight: 54.8 ± 5.3 kg) who were high performance at the international and national level. The BC (measured by dual-energy X-ray absorptiometry) parameters included body height and mass, body fat percentage (BF %), lean body mass (LBM), body mass index (BMI), total bone mineral density (BMD total), and total bone mineral content (BMC total). Training volume and pattern were assessed via an author-developed questionnaire, while injury status (i.e., anatomical location, symmetry and rate of injuries) was assessed via a modified questionnaire on overuse injuries used in sports injury epidemiology studies with elite-level athletes. Body composition parameters were as follows: the BMI was 21.5 ± 1.4 kg/m^2^, BF % was 21.9 ± 4.7%, LBM was 41.2 ± 3.4 kg, BMD total was 1.22 ± 0.08 g/cm^2^, and BMC total was 2486 ± 344 g. Furthermore, the average weekly volume of training was 23.5 ± 1.4 h, with the highest training volume occurring on Monday and Tuesday and high variability within gymnasts. Our study also revealed that the most frequently injured joints that had the most negative impact on training volume, sport performance and pain status were the ankles and low back, followed by the knees and shoulders. This kind of yearly screening method is warranted to allow more definitive conclusions to be made on adjusted training and preventive strategies.

## 1. Introduction

Body composition (BC) is used as one tool among several to ensure optimal preparation for competition. Body composition can be used as a guideline for weight management recommendations and concerns in competitive sports. The two main modifiable factors that directly influence body composition are (i) overall physical activity (i.e., regular activity and repeated training-related activity) and (ii) dietary intake (i.e., energy and nutrient intake) [1,2,3]. One of the sports in which BC is known to be a strong determinant of performance is female artistic gymnastics [4,5].

Female gymnasts participating in this sport experience many difficulties regarding their BC and visual body image [6,7,8]. The monitoring of BC is important for female athletes in general, but it is especially important for artistic athletes, as BC is associated with their social body image [6,7]. Not surprisingly, many studies have suggested that a lower body fat percentage (BF %) and an appropriate body image might be important contributors to success in artistic gymnastics [6,9,10,11,12]. Furthermore, a study among the most successful elite-level female gymnasts reported their BF % to be very low (ranging from 11.3 to 16%) [4]. Importantly, BC is assessed with numerous methods (i.e., bioimpedance, dual-energy X-ray absorptiometry (DEXA), skinfold measurement), so direct comparisons are oftentimes limited. Furthermore, many of the female gymnasts participating in studies are sub-elite-level and/or very young gymnasts but are characterized as elite-level athletes [4], so the results of recent studies including comparisons need to be interpreted with caution. Important, for the purpose of our research (e.g., for our gymnasts), we used the term *high performance level*.

Additionally, gymnasts commonly initiate systematic training at the age of six years, and concerns about body weight and BC are brought to their attention practically as soon as they are introduced to the sport [13]. On most days during the week, elite-level gymnasts usually have two practice sessions a day (one to four hours each), and they have one rest day per week. Typically, they train 20 h per week, while the peak season may require as many as 30 to 40 h of training per week [14,15]. Gymnasts are under pressure to optimize their BC status due to high expectations of judges in competitions, specialization required, and intensive training. The unique relationship with their coaches and the judges in the sport must be considered when suggesting solutions [16].

Athletes, especially gymnasts, constantly seek a balance between adequate nutrition, an appropriate BC, a favourable visual appearance, effective recovery and a good immune system [1,12,17,18]. However, it seems that the dietary intake of female artistic gymnasts may be nutritionally inadequate, and sometimes concerns about body weight lead to eating disorders (ED) [19,20,21]. Importantly, measures for preventing sport-related injuries should also be included in overall physical preparedness and training/dietary strategies. Of note, the incidence and severity of injuries is relatively high among artistic gymnasts, especially advanced-level female gymnasts [22,23,24]. A recent systematic review of 12 studies including 843 female gymnasts that evaluated injury type and frequency showed that most injuries affected the lower extremities, followed by the upper extremities, the torso/spine and neck [23].

There is a lack of recent data on various sport-related factors in high-performance-level artistic gymnasts, especially BC status during the competition period (i.e., most reports on these issues are relatively old [25,26]). This study is part of a larger cross-sectional study of the nutritional and cardiovascular health status of the high-performance-level gymnasts and swimmers (i.e., matched according to age and competition period) among these two indoor sports [27,28]. Thus, the aim of this study was to examine BC, training volume/pattern and injury status in high-performance-level female artistic gymnasts and make recommendations for the future health and well-being of these athletes, based on these finding and those existing in the literature.

## 2. Materials and Methods

### 2.1. Study Design and Eligibility

This study was approved on 28 March 2018 by the Slovenian Medical Ethics Committee (approval document no. 0120–177/2018) and registered at https://clinicaltrials.gov (approval document no. NCT03584256). All participants and/or the legal guardians signed an informed consent form for inclusion in the study. The participants were not compensated financially to participate in the study. The study was conducted on 4 April 2018 (during the competition period) within three hours. The BC and other measurements were performed at a certified biomedical centre (Ljubljana Medical Centre, Ljubljana, Slovenia) by their medical staff (biochemical assays) and a certified physician (BC measurements). All the assessments performed in the study were funded by the Slovenian Research Agency.

### 2.2. Subjects

We included 17 high-performance-level female artistic gymnasts from Slovenia, with a mean age of 17.4 ± 4.1 years. All the participants resided in Slovenia at a latitude of 46° N, and several of them were performing at the highest international competitive level (i.e., European Championships, World Cups, Olympic Games). The female gymnasts were recruited through personal contacts with the National Team Coach and other gymnastic coaches working in the clubs around Slovenia. The criteria for inclusion in the study were as follows: competing at the international and national level, currently fully actively involved in a training programme, and not using any prescribed medication that affects bone metabolism. All the gymnasts who were recruited participated in the study, and none of them were excluded from the study analysis. 

### 2.3. Outcome

The variables in this study included the basic characteristics of the athletes, anthropometric and BC parameters, training parameters (i.e., volume/pattern), and the injury pattern (i.e., location, symmetry and rate). Of note, dietary intake, serum micronutrients and cardiovascular health status are summarized only to understand the obtained results and are presented in more detail in separate manuscript [28].

#### 2.3.1. Characteristics of the Participants

The characteristics of the study participants (i.e., education, competition level, type of diet, menstrual status and characteristics) and the training status (i.e., beginning training, volume/pattern) were evaluated by a questionnaire developed by the authors.

#### 2.3.2. Anthropometric and Body Composition Parameters

The basic anthropometric parameters included body height, body mass, and body mass index (BMI), all of which were measured by experienced technicians using standardized medical approved professional personal floor scale with stand (Kern, MPE 250K100HM, Kern & Sohn, Balingen, Germany) and the corresponding protocols (i.e., body mass index was calculated as weight in kilograms divided by the square of height in metres).

The body composition (BC) measures included body fat percentage (BF %), lean body mass (LBM), total bone mineral density (BMD total), bone mineral density of the spine (BMD spine), bone mineral density in the left femoral neck (BMDLF neck), and total bone mineral content (BMC total). For the evaluation of the BC parameters, DEXA (General Electric Company, model Lunar Prodigy 5) with EnCore software, version 13.31, was used.

#### 2.3.3. Injury Status

The injury location, symmetry and the rate of injury were evaluated via a modified questionnaire based on a method of assessing overuse injuries used in sports injury epidemiology studies [29] and an approach to prospectively monitoring illnesses and injuries in high-performance-level athletes [30]. The questions concerned the period of the last six months. The questions were divided into four sections: (1) problems with participation in training and competitions due to problems with the selected joints, (2) to what extent gymnasts reduce training volume due to problems with the selected joints, (3) to what extent the problems with the selected joints affect sport performance, and (4) to what extent gymnasts feel pain due to problems with the selected joints in sports activities.

#### 2.3.4. Dietary Intake, Serum Micronutrients and Cardiovascular Health Status

We evaluated dietary and supplements intake using a standardized Food Frequency Questionnaire (FFQ), serum micronutrients (i.e., B_12_, 25-hydroxyvitamin D (25(OH)D), calcium, magnesium, phosphorus, potassium, and iron) as well as several cardiovascular health markers (i.e., blood lipids and blood pressure) using biochemical assays and blood pressure measurement. Nutritional and cardiovascular health information are provided in more detail in a separate article [28].

### 2.4. Statistical Analysis

Statistical analysis was performed with R 3.5.2 with the dplyr [31], ggplot2 [32] and arsenal [33] packages. Dplyr was used for data transformation, ggplot2 was used for data visualization, and arsenal was used for statistical tests. The number of high-performance-level athletes is extremely small; therefore, we were targeting all artistic gymnasts that met our inclusion criteria. All the gymnasts who were invited and met the criteria, accepted our invitation. No data were missing. No sensitivity analyses were performed. The data are presented as means ± standard deviations (SD). For non-continuous data, frequencies (number of observations–n) and percentages (%) are reported.

## 3. Results

### 3.1. Characteristics of the Gymnasts

The majority of the gymnasts had completed or were still attending high school at time of testing (65%). The participants started with regular gymnastics training at a mean age of 5.3 ± 2.7 years, and they had all been involved in systematic training for at least 7 years, with an average of 10.5 ± 3.4 years of systematic training. More details on the characteristics of the athletes are presented in Table 1.

### 3.2. Anthropometrics and Body Composition Status

The average body height, body mass, BMI, and BF % were 159.8 ± 6.2 cm, 54.8 ± 5.3 kg, 21.5 ± 1.4 kg/m^2^, and 21.9 ± 4.7%, respectively. Importantly, the mean BMI and BF % of the two gymnasts who competed at the highest level before the study (World Cup) were lower than the average BMI and BF % (i.e., 19.7 and 21.1 kg/m^2^ and 16.3% and 20.8%). More details on the BC status of the gymnasts are presented in Table 2.

### 3.3. Training Volume and Pattern

The gymnasts reported training for an average of 23.5 h per week. The days on which the highest total training volume occurred for the gymnasts were Monday (4.0 h/day), Tuesday (3.9 h/day), and Friday (4.0 h/day); for the remaining days, the training volume was modestly lower on Wednesday (3.6 h/day) and was similar on Thursday (3.7 h/day) and Saturday (3.6 h/day). Sunday was mostly a rest day (0.7 h/day). However, 29% of gymnasts practised on Sunday; the duration of a single workout was two hours long, and the session primarily consisted of recovery or prevention training. Interestingly, at the time of the study, two of the most successful gymnasts reported practising 20 (specialist on two apparatus) and 32 h per week (all-round specialist). In addition, for the most successful of these two gymnasts, the practice volume on Monday, Tuesday, Thursday, and Friday was six hours daily, and on Wednesday and Saturday was four hours daily, and Sunday was a rest day.

### 3.4. Injury Status

The areas of the body that were most commonly injured were the ankles and low back. Ankle and low back injuries limited the volume of gymnast training the most, while knee injuries decreased the training volume and negatively affected sport performance the most. Furthermore, the gymnasts felt moderate or severe pain in the ankles and lower back, followed by the knees and elbows. Due to ankle injuries, 35% of gymnasts had to limit training, and the second most-limiting injuries were low back injuries (23%). Importantly, 41% of the gymnasts decreased their training volume due to low back or shoulder injuries. A total of 41% of the gymnasts reported that pain in their low back (80% of cases were symmetrical) or knees (75% were not distributed symmetrically) was the most common reason for a decrease in performance within the last 6 months. Moderate and severe pain were the most frequently reported severities of pain in the ankles (25% more cases on the right ankle) and low back (11% more cases on the right side). Injury assessment is presented in Table 3. 

### 3.5. Dietary Intake, Serum Micronutrients and Cardiovascular Health Status

The mean (± SD) energy intake was 1514 ± 258 kcal/day (i.e., the dietary supplements and sport drinks were included in the evaluation of dietary intake). Dietary supplements or sport drinks were consumed by 59% of gymnasts (i.e., the most frequently consumed dietary supplement was magnesium (35%)). Dietary intakes from foods and supplements are presented in Table 4. Further, the most challenging serum micronutrient shortcomings were 25(OH)D (77%) and vitamin B_12_ (47%). Finally, all cardiovascular and safety variables (i.e., lipids and blood pressure, serum fasting glucose, uric acid, and haemoglobin) were within recommended ranges; however, 41.2% of them had low-density lipoprotein cholesterol above 3 mmol/L and 35% had triglycerides ≤0.6 mmol/L [28].

## 4. Discussion

### 4.1. Main Findings

In this study of the BC, training parameters and injury status of high-performance-level female artistic gymnasts, there were three important findings. First, the gymnasts’ BC differed from those found in previous studies and recent reviews. Second, training volume varied, even within the most successful gymnasts, but the average total training volume in the competition period was consistent with those reported in other studies. Third, the areas of the body that were the most susceptible to injuries and when injured, had the largest negative impact on training, training volume, sport performance, and pain (moderate or severe) were the ankles, low back, and knees.

### 4.2. Anthropometrics and Body Composition Status

Mean BC in our sample was higher than other studies that reported on elite- or high-performance-level female athletes in artistic gymnastics [4,26]. The reason might be suboptimal dietary intake [28] and interseason changes in BC, which are known to occur among team sport athletes [35] and among female gymnasts [36,37]. Although this is based on two study participants, we observed that mean BF % of the gymnasts who successfully participated in the most elite competition just before the study (World Cup) was lower than average (e.g., 16.3% and 20.8% compared with 21.9%). Unfortunately, we did not measure the BC of these two gymnasts before the competition period. 

In a previous cross-sectional study on six artistic gymnasts, members of the Danish National team (aged 17.9 vs. 17.4 years in our study) had a higher mean body height than that found in our study (165 vs. 159.8 cm), a similar body mass (53.7 vs. 54.8 kg), a lower BMI (19.7 vs. 21.5 kg/m^2^), a higher relative LM (81.7 vs. 75.2%), and a lower BF % (14.1 vs. 21.9%) [38]. The researchers who conducted another cross-sectional study on 48 younger artistic gymnasts (15.2 years), members of the US National team, found a significantly lower BF % than we did (14.3%) [26]. Interestingly, the gymnasts with the highest BF % in the US study (21.5%) still had lower BF % than was estimated as the average in our study. 

The time in the season and the level of performance are likely very important considerations when comparing results of our study to previous results. For example, higher level of performance has been associated with a lower BF % [25,26,36,37]. However, several studies have shown that female gymnasts have difficulties achieving the optimal BC status [4,25,26]. Furthermore, the extent to which a lower BF % impacts performance in high-performance-level gymnasts remains controversial. Many female gymnasts included in studies are sub-elite-level athletes and/or very young gymnasts but are classified by researchers as elite-level or highly competitive athletes. Therefore, the results of studies including comparisons need to be interpreted with caution [4]. For example, in a recent review of the physique of female artistic gymnasts, researchers reported body height to range from 158–163 cm, body weight to range from 51.5–57.9 kg and BF % to range from 12.3 to 19.5% (age range was 15.1–15.7 years) [4]. Importantly, energy deficits were found to be associated with a higher BF % [25]. In brief, a previous study reported that 75% of female gymnasts admitted to using inadequate weight loss strategies [39] (i.e., female gymnasts are subject to pressure to be thin, and retain of body mass at low level [40]), while newer studies have supported these findings, where the researchers found that 65% of the surveyed gymnasts had asymptomatic ED and almost 30% had subclinical ED [19]; therefore, female gymnasts should be educated on how to avoid using unhealthy dietary practices, achieve their optimal BC and prevent suboptimal after-training recovery [19,39]. To conclude, at this point it is difficult to say where exactly the optimal BF % values/ranges are, so our assessment of the results obtained (that the values we reported are higher than those in the literature) is based on the comparison with the available data. The reasons for this lie in differences in the comparative age of gymnasts in other studies, different methods used to assess body composition status, the vagueness of terms *elite level* or *high performance level* used, and the associated level of competition, the high proportion of ED that may lead to lower BF %, differences in motor skills requirements in case the gymnast competes in a single discipline (e.g., in floor or vault exercises, explosive power is more pronounced that aesthetic) or among all-round specialists. Lastly, female artistic gymnastics is not a sport of maximum effort. Body control is a more important component and consequently there is a possibility to compensate certain physical potentials with other abilities.

Furthermore, the BMD of the gymnasts in our study was somewhat higher than that of the 48 members of the National US team measured in the 2005 study mentioned above with respect to BF % (1.22 and 1.06 g/cm^2^ for Slovenian and US gymnasts) [26]. Importantly, this inconsistency may also be a result of the participants in the current study being older (17.4 and 15.2 years for Slovenian and US gymnasts), taller (159.8 and 152.2 cm for Slovenian and US gymnasts) and heavier in terms of body mass (54.8 and 47.7 kg for Slovenian and US gymnasts). Contrary to these results, an older Danish study in six elite-level artistic gymnasts, with which we compared our anthropometric and BF % results above, reported a higher mean BMDLF status than we did in 17 high-performance-level artistic gymnasts (1.33 g/cm^2^ vs. 1.24 g/cm^2^) [38]. When we compared our BMDLF values at the time of the study for the five most successful gymnasts (1.27 g/cm^2^), we still found lower BMDLF values in our study than in the Danish study. The reasons for this inconsistency might be a suboptimal 25(OH)D level and dietary status among the gymnasts in our study [28].

### 4.3. Training Volume/Pattern

The expected training volume/pattern was observed; the training volume was higher on Monday, Tuesday, and Friday, and on the other days (Wednesday and Saturday), training volume decreased slightly; Sunday was mostly a rest day. However, some gymnasts (29%) had a short training session on Sunday, but the session consisted of mostly recovery- or prevention-type training. Importantly, at the time of the study, the most successful gymnast (i.e., all-round specialist) performed 32 h of training per week, while the next most successful gymnasts (i.e., specialist for two apparatus) performed significantly lower training volumes (e.g., 20 h per week). These data indicate a relatively large and important inter-individual variability in training volume as well as individual conditioning training tapering during the competition period for training periodization.

It is well established that a typical amount of training for elite-level gymnasts ranges from 20 to 30–40 h of training per week [14,15]. The gymnasts are learning a wide range of skills and techniques, so they should have an extremely high level of conditioning preparedness. Due to repetitive training, gymnasts usually practice in a constant state of fatigue [41,42], and due to the high training volume, they may not recover completely between training workouts. High expectations in gymnastics environments and specialization, daily intensive repeated trainings that begin at an extremely early age may contribute to the constant pressure to optimize one’s body composition [16] and lead female artistic gymnasts to have an inadequate dietary intake [20,28,43,44,45] suggests that dietary intake still needs to be optimized to support the volume, intensity and frequency of training. Furthermore, one possible reason for obtained higher BF % status of our participants might be due to their low protein (14% of calories) and high fat intake (40% of calories), and as we reported in our separate article, and due to high free sugar intake (17% of calories) [28]. Abiding by the recommendations (based on dietary energy density principles) enables better hunger management and promotes healthy body weight [46], and at the same time also addresses a complex behavioural phenotype (e.g., the loss of control over eating) [47].

### 4.4. Injury Status

In our study, we found that the most injured areas of the body were the ankles and back, followed by the knees. These three areas also had the most negative impact on training, training volume, and sport performance and led to moderate or severe pain. Overall, when we take into account all the measured variables, our results showed that the low back was as injury sensitive as the ankles.

The high and repetitive intensity of practice and competitions, the large number of events, and the technical difficulties make gymnastics one of the most injury-prone sports [48]. Our results are partially in line with those of the two review studies on injury type and frequency in gymnastics [22,23]; the authors found that most injuries affected the lower and upper extremities, followed by the back, shoulders and neck. Another recent review of 10 studies that investigated the location of injuries in female gymnasts reported the lower limbs were the most commonly injured area of the body [24]. However, older results in 64 Australian elite-level and sub-elite-level female gymnasts showed that the low back was the second most commonly reported area of injury, immediately after the ankle and foot [49]. Additionally, the researchers found that elite-level gymnasts spent 21% of the year training at less than full capacity due to injuries. Therefore, Australian researchers suggest several important measures to prevent injuries in elite-level gymnasts, such as ensuring a healthy nutritional diet and good mental health, sleeping well and making rest mandatory, designing the proper training loads, and using protective equipment [16]. Interestingly, a recent cross-sectional study in 63 skilled athletes, of whom 21 were female artistic gymnasts, showed that the most common cause of injury reported by the athletes was overload, followed by poor techniques and incorrect training methodologies. Importantly, the researchers in their self-assessment questionnaire did not include nutrition as a possible external factor [50].

There is various evidence on how to effectively reduce the rate and severity of gymnast injuries while maximizing gymnastic performance, including implementing appropriate training modalities that include mental training to enhance self-esteem, maintaining the appropriate BC, maintaining adequate nutrition, using protective equipment, and spotting [23,24,51,52]. Importantly, an appropriate nutritional status evidently supports repeated training adaptation, reduces the risk of injuries and illnesses, increases the possibility for athletes to perform as expected and supports optimal health [53]. However, individual dietary requirements are influenced by a variety of known factors, such as age, sex, and body mass, as well as by sport-specific factors, such as the type of sport, training volume and intensity, periodization phase, and phase within the competition period [54,55]. Furthermore, most of the limited number of studies on female gymnasts do not support the notion that female gymnasts’ diets are nutritionally adequate [20,43,44,45]. In line with this observation, gymnasts in our study (published in separate article) have room for dietary optimization which would also benefit their training, athletic performance, and health status [28]. Based on the examined dietary intake of our participants, dietary recommendations may include an increased intake of whole plant-based foods (especially whole grains, legumes, fresh fruits and vegetables and moderate intake of nuts/seeds) as well as small sea fish (sardines, sardines, and anchovies), and limited quantities of lean meat, low-fat dairy products and liquid vegetable oils, while decreasing the intake of processed, ultra-processed and fried foods [56,57].

## 5. Limitations and Future Directions

The strength of our study was that we assessed all high-performance-level gymnasts from Slovenia, where gymnastics is commonly practiced, with many Olympic and world-class gymnasts being from Slovenia over the last 100 years [58]. For BC assessment, the use of DEXA combined with other broad measurements already published (i.e., dietary intake status and extensive blood assay (i.e., serum micronutrients, cardiovascular health and safety status)) [27,28], added value to these findings. Furthermore, all the assessments were performed on the same day, in the same certified medical centre, following the same methods, after an overnight fasting state, and within a three-hour period. Given the general lack of up-to-date data on high-performance-level artistic gymnasts, which would allow for a greater validity of recommendations about the desired body composition status, we see our data as a relevant contribution to the research area. 

However, the study has known limitations inherent to the nature of cross-sectional studies and the sample size of the high-performance-level gymnasts. Therefore, due to these limitations, the results should be interpreted with caution, replicate studies with larger sample sizes should be performed. Importantly, the questionnaires that characterized the participants (i.e., education, competition level, type of diet, menstrual status and other characteristics) and their training status (i.e., beginning training, volume/pattern) were self-reported; and reliability and validity were not estimated for these questionnaires; therefore, further replication is needed. Only the volume and weekly training pattern were reported, while the intensity and content of each training session were not assessed. Similarly, only anatomical location and the rate of injury were assessed, and the injury type (i.e., sprains, strains, fracture, and pain) and cause (i.e., acute, overuse, and safety in training) were not reported, which may change how the injury data is interpreted. Our study also did not analyse motor skills (e.g., explosive strength and power) or neuromuscular fatigue that can affect performance. Future research using longitudinal research should also address the question about the relationship between various interrelated parameters such as training volume, dietary intake, BC, injuries and sport performance/success.

## 6. Conclusions

The results demonstrated that the average BF % status was higher than found in most other studies on elite or high-performance-level female artistic gymnasts. Additionally, we found that the volume of training varied within gymnasts, even among the most successful ones, and should be viewed as an individual characteristic; training differs in response to individuals’ conditions, the competition period and their recovery as well as dietary support. However, the average total weekly training volume was found to be in line with the results reported in previous studies for the competition period. Similarly, in line with the results of most other studies, the anatomical location and the injury rate evaluation showed that the ankles, low back and knees are the most susceptible areas to injury, have the most negative impact on training volume and sport performance, and led to moderate or severe pain. 

Female gymnasts, coaches, researchers, and the gymnastic system urgently need to effectively solve the long-standing challenge of achieving and maintaining the desired BC, a healthy body image, and effective post-training recovery and improving long-term injury prevention, thereby improving gymnasts’ sport performance and health. Taken together, our findings call for periodic screening and potentially for adjusted strategies (i.e., training, safety, spotting, and nutrition) to improve BC status and to decrease injury rates. 

## Figures and Tables

**Table 1 ijerph-18-02019-t001:** Characteristics of the study participants (mean ± SD).

Parameter	N = 17
Age (years)	17.4 ± 4.1
*Training* (*years*)	10.5 ± 3.4
Beginning training gymnastics (years)	5.3 ± 2.7
Weekly training volume (hours/week) ^†^	23.5 ± 3.4
*Education* (*n* (%))	
Elementary school (attending)	6 (35)
High school/Gymnasium (attending/completed)	11 (65)
*Competition level* (*n* (%)) ^††^	
EC or WC or OG medallist/finalist/attendee	5 (29)
NC medallist/finalist/attendee	12 (71)
*Type of diet* (*n* (%)) ^‡^	
Vegetarians	3 (18)
Occasionally vegetarians	5 (29)
Omnivores	9 (53)
*Menstrual status* (*n* (%))	
Post-menarcheal	15 (88)
Pre-menarcheal	2 (12)
First time (years)	13.9 ± 1
*Menstrual characteristics* (*n* (%))	
Regular	14 (82)
Not regular ^‡‡^	3 (18)
Painful	5 (33)
Not painful	10 (67)

^†^ The training volume was a reflection of the competition period over the study period and the type of the specialization (e.g., all-round and individual apparatus specialists). ^††^ Competition level: EC (European Championship), WC (World Cup), OG (Olympic Games) and NC (National Championship). ^‡^ Dietary intake was assessed using a validated food frequency questionnaire (FFQ) that allowed us to distinguish subpopulations with different dietary patterns [28]. ^‡‡^ Not regular (menstrual cycle) category includes two gymnasts who had not yet had the onset of menstrual cycle.

**Table 2 ijerph-18-02019-t002:** Anthropometrics and body composition status (mean ± SD).

Parameter	N = 17
Body height (cm)	159.8 ± 6.2
Body mass (kg)	54.8 ± 5.3
BMI (kg/m^2^)	21.5 ± 1.4
BF %	21.9 ± 4.7
LBM (kg)	41.2 ± 3.4
BMD total (g/cm^2^)	1.22 ± 0.08
BMDLF neck (g/cm^2^)	1.24 ± 0.11
BMD spine (g/cm^2^)	1.09 ± 0.09
BMC total (g)	2486 ± 344

BMI: (body mass index), BF %: body fat percentage, LBM: lean body mass, BMD total: total bone mineral density, BMD spine: bone mineral density of the spine, BMDLF neck: bone mineral density in the left femoral neck, BMC total: total bone mineral content.

**Table 3 ijerph-18-02019-t003:** Injury assessment.

Parameter ^†^	Ankles	Knees	Hips	Back ^††^	Shoulders	Elbows	Wrists	Fingers
*Training/competition problems* (*n*)								
Executed training with little/no health problems	12	17	17	14	17	17	16	16
Limited training	6	3	1	40	1	1	3	2
No training	1	1	1	1	1	0	1	1
*Training volume* (*n*)								
Decreased training volume	5	5	1	7	7	1	4	1
Limited training	5	4	0	5	3	0	2	2
No training	0	0	1	0	0	1	1	0
*Influence on sport performance* (*n*)								
No influence	11	12	13	9	11	13	13	14
Limited performance	5	7	2	7	2	3	2	2
No performance	1	0	0	1	0	0	1	0
*Pain status* (*n*)								
No or mild pain	11	12	15	10	16	10	11	13
Moderate/severe pain	7	4	0	6	0	4	3	2

^†^ The rate of injury for the particular parameter is reported as the number of gymnasts (*n*). ^††^ Back: low back region.

**Table 4 ijerph-18-02019-t004:** Dietary intake (mean ± SD).

Variable	N = 17
**Dietary intake**	
Energy intake (kcal/day)	1514 ± 258
Carbohydrates (g)	177 ± 39
(% E)	47 ± 11
Dietary fibre (g)	11 ± 2
Total fat (g)	67 ± 24
(% E)	40 ± 13
SFA (g)	30 ± 13
(% E)	18 ± 7
Cholesterol (mg)	148 ± 63
Protein (g)	49 ± 12
(% E)	14 ± 3
Water (L) ^TW^	1.97 ± 0.21

Data are presented as the means (standard deviation). Atwater energy conversion factors were used (kcal/g): carbohydrates and protein = 4, dietary fibre = 2, fat = 9 [34]. ^TW^ Total water from solid foods and beverages (including sports drinks).

## Data Availability

The data used to support the findings of this study are included within the article.

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
