# Peer review of "Body Composition, Training Volume/Pattern and Injury Status of Slovenian Adolescent Female High-Performance Gymnasts"

_ijerph, 2021, doi:10.3390/ijerph18042019_

Round 1

Reviewer 1 Report

General Comments

Thank you very much for giving me the opportunity to review your work. The topic of the paper is interesting and fits the scope of the journal. The text itself is well written and composed. However, there are some methodological issues that need to be clarified.

Specific Comments

L 93: Please consider providing more information regarding the rationale behind sample size selection. Do you believe that 17 females are enough to generalize your results? If possible, a power analysis could answer this question.

L 111-112: Please provide more information regarding the procedures you followed to examine the validity and reliability of the questionnaire you developed.

L 115-123: Since your article is heavily depended on this body composition assessment, it is of great importance to provide more information regarding all the protocols followed. For example, it would be essential to provide information on important parameters, such as the hydration status of the measured athletes, that are able to impair the assessment of body composition.

L 187-189: It would be of great importance to conduct appropriate statistical analyses describing this relationship between the hours of training and their success. Moreover, the same analysis could be extended to examine similar relationships between the prevalence of injuries, body composition, and all the variables examined in the sections 3.1 to 3.5.

Reviewer 2 Report

This is an overall well-written and scientifically sound manuscript. Methodology for body composition assessment is robust, and outcomes are appropriately reported. Much of what this study reports has been reported previously in similar populations; because of this, the novelty and significance are not as great as if additional outcomes were assessed that were not previously reported. However, this study has merit for comparison of elite gymnasts in differ parts of the world.

One conclusion that was not clear to me was the statement that the average BC status is potentially suboptimal. This is mentioned both on line 232 and on lines 365-366. In the conclusion, the authors go on to state that the extent of influence of a lower or higher BF% on sport performance remains controversial. So, I am not clear if "suboptimal" is meant to imply that BF% would be more optimal if it was lower or higher. Because it was discussed that other studies report lower BF% in similar populations, my guess is that the authors mean that BF% would be more optimal at a lower %. But, this is not clear. I recommend revising the discussion to clarify what is meant by the statement that BC status may be suboptimal.

Minor revisions suggested:  Section 2.1, Paragraph 1, has a redundant sentence (lines 85 & 89). Correct minor typos and grammatical throughout the manuscript (e.g., line 140, detailed should be present tense and I would consider rewording the last part of this sentence).

Reviewer 3 Report

Manuscript ID: ijerph-1091954

“Body Composition, Training Volume/Pattern and Injury Status of Slovenian Adolescent Female High Performance Gymnasts”

Level of interest: I have revised this manuscript, this interesting article should be accepted for publication.

Quality of written English: Well done

Statistical review: Right

Declaration of competing interests: I have no competing interests to declare, hold no shares.

“Body Composition, Training Volume/Pattern and Injury Status of Slo- 2 venian Adolescent Female High Performance Gymnasts”

The authors state that the aim of the submitted manuscript was to evaluate body composition (BC), training volume/pattern and injury status in high performance female artistic gymnasts. This is an interesting and worthwhile cross-sectional study of 17 subjects and the authors have put in a good effort on the project. Moreover, the research in my opinion is well written.

However, there are several details that could be improved on this manuscript.

1) INTRODUCTION: Line 38 to 539 on page 1: Please, could you add the reference in these lines?

2) *Line 56 to 61 on Page 2. Please, I would recommend you that you can rewrtite this paragraph. From my opinion, It is too long.

3) METHODS: The authors should clarify this fact. This study only considers 17 subjects. It does not seem to be a high amount.

4) Rephrasing is needed in the following area to enhance clarity:

Line 83 to 91 on page 2, please, could you reduce and improve the wording in these lines?

5) Line 112 on page 3: Please, could you explain this questionnaire and add an appendix in this section?

6)RESULTS: For instance, regarding to dietary intake, it would be of great help for the readers to add a table synthesizing clearly these results, This table should include on result section.

7) DISCUSSION: Line 248 to 60 on Page 7. Please, I would recommend you that you can rewrtite this paragraph. From my opinion, It is too long.

It comes to my attention that the authors do not analyze / synthetize / discuss anything about other outcomes such as strength. Please, I recommend you that you add in your limitations.

8)The conclusions are well written.
